# Control over the Surface Properties of Zinc Oxide Powders via Combining Mechanical, Electron Beam, and Thermal Processing

**DOI:** 10.3390/nano12111924

**Published:** 2022-06-04

**Authors:** Igor A. Pronin, Igor A. Averin, Andrey A. Karmanov, Nadezhda D. Yakushova, Alexey S. Komolov, Eleonora F. Lazneva, Maxim M. Sychev, Vyacheslav A. Moshnikov, Ghenadii Korotcenkov

**Affiliations:** 1Department of Nano- and Microelectronics, Penza State University, 440026 Penza, Russia; nano-micro@mail.ru (I.A.A.); starosta07km1@mail.ru (A.A.K.); yand93@mail.ru (N.D.Y.); 2Solid State Electronics Department, St. Petersburg State University, 199034 St. Petersburg, Russia; a.komolov@spbu.ru (A.S.K.); e.lazneva@spbu.ru (E.F.L.); 3Department of Theory of Materials Science, Saint-Petersburg State Institute of Technology (Technical University), 190013 St. Petersburg, Russia; msychov@yahoo.com; 4Department of Micro- and Nanoelectronics, St. Petersburg State Electrotechnical University, 197376 St. Petersburg, Russia; vamoshnikov@mail.ru; 5Department of Physics and Engineering, Moldova State University, 2009 Chisinau, Moldova; ghkoro@yahoo.com

**Keywords:** zinc oxide, electron beam (e-beam) irradiation, surface, X-ray photoelectron spectroscopy

## Abstract

The surface properties of zinc oxide powders prepared using mechanical activation, electron beam irradiation, and vacuum annealing, as well using combinations of these types of treatments, were studied using X-ray photoelectron spectroscopy. The structure of the obtained materials was studied by an X-ray diffraction technique and by scanning electron microscopy. We found that over five hours of grinding in an attritor, the size of nanocrystals decreases from 37 to 21 nm, and microdeformations increase from 0.3% to 0.6%. It was also found that a five-hour grinding treatment promoted formation of vacancies in the zinc sublattice at the surface and diffusion of Zn^2+^ cations into the bulk of the material. Irradiation of commercial zinc oxide powders with an electron beam with an energy of 0.9 MeV and a dose of 1 MGy induced breaking of Zn–O bonds, diffusion of interstitial zinc ions into the bulk, and oxygen atom escape from regular positions into the gas phase. A combined treatment of five hours of grinding and electron beam irradiation promoted accumulation of interstitial zinc ions at the surface of the material. Annealing of both initial and mechanically activated ZnO powders at temperatures up to 400 °C did not lead to a significant change in the properties of the samples. Upon exceeding the 400 °C annealing temperature the X-ray photoelectron spectra show almost identical atomic composition of the two types of materials, which is related to diffusion of interstitial zinc ions from the bulk of the material to the surface.

## 1. Introduction

At present, zinc oxide (ZnO) is one of the most commonly used materials for application in gas sensors, catalysts and photocatalysts, phosphors, and piezoelectric and solar cells [1,2]. Regarding optimal parameters, metal oxides are known to have a uniform size of small nanocrystals and the required concentration of point defects that determine the surface activity. This is usually realized by using various physical and chemical exposure methods [3,4]. For example, dispersion of commercial or synthesized powders is a method for optimizing the structure of the used metal oxide, carried out by particle size reduction, changing the porosity, and improving the morphology of the formed active layer. In particular, the influence of milling conditions (speed, time, medium) on the sensory properties of ZnO powders has been investigated, and the optimal dispersion parameters with maximum response in various gas environments have been identified in [5]. Similar studies have been carried out for other ZnO applications [6,7]. However, dispersion cannot have a selective exposure effect on point defects. In this regard, an approach using electron beam (e-beam) irradiation is very attractive. According to [8], irradiation of ZnO nanofibers with e-beam energy of 1 MeV at 150 kGy dosage allows improvement of the selective properties in hydrogen gas detection, and a considerable increase in the sensor response. As stated in [9], having undergone e-beam irradiation, europium-doped thin ZnO films show potential for applications as a source of red light. The possibility of adaptable modification of magneto-transport characteristics of cobalt-doped ZnO powders due to e-beam irradiation has been reported [10,11].

Despite the fact that the effects of both mechanical milling and e-beam irradiation on surface properties of ZnO powders have been well studied, we provide information on their cumulative impact for the first time. We have found that commercial and dispersed ZnO powders have different interaction mechanisms with the beam, which leads to various combinations of point defects in the boundary layer. In addition, some combinations can only be realized with successive powder milling and irradiation. In this study, using the technique of X-ray photoelectron spectroscopy (XPS), we have studied both the effects of mechanical milling, ultra-high vacuum (UHV) annealing and e-beam irradiation, and some combinations thereof on the surface properties of commercial ZnO powders. Structural parameters of the samples have been evaluated using electron microscopy (EM) and X-ray phase analysis.

## 2. Materials and Methods

### 2.1. Sample Preparation

Dispersion of commercial ZnO powders (pure for analysis, CJSC Vekton, St. Petersburg, Russia) was carried out in an attritor (type: BATCH-LAB, model: HD/01, Union Process, Inc., Akron, OH, USA) using a 2 L corundum crucible and zirconium oxide (ZrO_2_) grinding balls with a diameter of 3 mm. The ball-to-powder mass ratio is 18:1. Milling was performed in an isopropyl alcohol medium under the following conditions: milling speed is 400 rpm; milling time is 5 h. The resulting powder was exposed to air-drying at a temperature of 70 °C for 24 h. No other chemical treatment was implemented.

Irradiation of sample surfaces of commercial and dispersed ZnO powders was performed with e-beam energy of 0.9 MeV at 1 MGy dosage for 5 h using an RTE-1V resonant transformer electron accelerator.

Annealing of commercial and dispersed ZnO powders was carried out under UHV conditions (10^−7^ Pa) in the temperature range of 200–500 °C for 1 h using an X-ray photoelectron spectrometer.

### 2.2. Sample Study

The surface structure of samples was investigated using a VEGA 3 SBH scanning electron microscope (TESCAN, Brno, Czech Republic) with a reflected electron detector.

The crystal structure from powder diffraction was determined using a DRON-3M automatic X-ray diffractometer (JSC BOUREVESTNIK, St. Petersburg, Russia) with CoK_α_ radiation in the range of 35° < 2*Θ* < 85°. Based on the diffractograms obtained, the sizes of the coherent scattering regions and microstrain were determined as follows. The broadening of the diffractogram reflection at half its height *B* is a mathematical convolution of the broadening associated with the polycrystalline state of the sample (coherent scattering regions) *B_D_* and with the broadening associated with the presence of microdeformations *B_S_*. These values are given by the following formulas. *B_D_* = *Kλ*/(*D*cos*Θ*), *B_S_* = 4*η*tg(*Θ*), where *K* is the form factor; *λ* is the wavelength of X-ray radiation; *D* is the size of the coherent scattering regions; *Θ* is the diffraction angle; *η* = Δ*a*/*a* is the value of microstrains, defined as a relative change in the lattice period *a*. If one considers a reflex profile as a Cauchy distribution, then *B* = *B_D_* + *B_S_*. Analyzing the broadening of several reflections (in principle, two are sufficient), *D* and *η* may be calculated. This procedure is discussed in more detail in Ref. [12].

The chemical composition of the surface of the obtained samples was analyzed using XPS. The XPS spectra were measured under UHV conditions (10^−7^ Pa) using an Escalab 250Xi X-ray photoelectron spectrometer (Thermo Fisher Scientific Inc., Waltham, MA, USA) with photon energy Al-Kα = 1486 eV. The XPS peak deconvolution was carried out by means of a Shirley background subtraction followed by peak fitting to Voigt functions having a mixed Gaussian and Lorentzian character. In order to remove the surface contamination related to atmospheric adsorbates [13], the surface of ZnO was softly etched by Ar^+^ ions (I_i_ = 1 μA, t_i_ = 30 s). The energy scale of the spectrometer was calibrated using a sputter-cleaned Au surface as a reference so that the binding energy of Au_4f7/2_ peak was set to 84.0 eV.

## 3. Results and Discussion

### 3.1. Scanning Electron Microscope and X-ray Diffraction

Figure 1 shows scanning electron microscopy (SEM) micrographs of the initial ZnO powders and those mechanically activated for 5 h. An analysis of the images has evidenced a decreased average particle size of the powder and an increased monodispersity thereof due to 5 h of milling. It can be seen that there are both large particles of about 1 μm and small ones of about 100–200 nm in the original samples. The milling has resulted in dissolution of large particles, while the powders have acquired the structure of elongated prisms with a base dimension of about 100 nm. E-beam irradiation of both initial and mechanically activated powders has not led to major changes in their structure.

Diffraction patterns of the samples investigated are presented in Figure 2. According to our X-ray diffraction (XRD) results, we have found a decrease in the coherent scattering region size from (37 ± 1) to (21 ± 1) nm, and an increase in microdeformations from (0.3 ± 10%) to (0.6 ± 10%)% during 5 h of milling. Irradiation of the samples with an electron beam had no significant effect on the obtained diffraction patterns. Similar results are presented in Ref. [14].

### 3.2. X-ray Photoelectron Spectroscopy

Figure 3a,b show the XPS spectra of Zn_2p_ and O_1s_ calibrated by taking the C_1s_ peak (284.8 eV) as reference for all types of material processing: 1—initial commercial ZnO powders; 2—initial e-beam-irradiated powders; 3—powders milled for 5 h; 4—e-beam-irradiated powders milled for 5 h [15].

It can be seen from Figure 3 that the spectrum of Zn_2p3/2_ shows symmetrical features and can be represented by a single component of Zn(*lat*.) with lattice parameters for all processing types, except for the fourth one [16]. On the contrary, the oxygen spectrum shows asymmetrical features in all cases and can be represented by a superposition of two components: a low-energy component with a binding energy of ~530 eV attributed to O^2−^ anion (O(*lat*.)) in the ZnO crystal lattice, and a high-energy component with a binding energy of ~531.5 eV attributed to oxygen in the composition of various particles adsorbed onto the surface of powders (O(*ads*.)) [16]. An analysis of the obtained data has shown that 5 h of milling of ZnO powder leads to an increase in the binding energy of Zn_2p3/2_ and O_1s_, while e-beam irradiation of both the initial and milled powders leads to a decrease in the binding energy of both peaks. Let us consider the obtained results in more detail. Table 1 shows the values of the binding energy of Zn_2p3/2_, O_1s_, and C_1s_ for all types of powder processing. It should be noted that an increase in the binding energy of Zn_2p3/2_ is associated with an increase in Zn oxidation state in the crystal lattice, while a decrease thereof is associated with a decrease in Zn oxidation state [17]. In this case, the key process related to an increase in the binding energy is the transition of Zn cations from a regular position (Zn_Zn_) to an interstitial site (Zn_i_) with the formation of Zn vacancy (V_Zn_) [18]:Zn_Zn_ → Zn_i_ + V_Zn_.(1)

In the case of Zn_i_ diffusion into the material bulk, there is a change in the surface stoichiometry with a decrease in the value of [Zn]/[O]. Another possible mechanism is the capture of Zn^2+^ cations by grinding media made of ZrO_2_.

A decrease in the binding energy of Zn_2p3/2_ is related to the release of the lattice oxygen from a regular position (O_O_) with the formation of oxygen vacancy (V_O_) [16]. In this case, possible transition mechanisms can be described by the following quasi-chemical equations:O_O_ → V_O_ + ½O_2_(*gas*)↑,(2)
O_O_ → V_O_ + O_i_,(3)
where O_i_ is interstitial oxygen. In this case, Process (2) is accompanied by a change in the near-surface stoichiometry. It should be noted that with the predominance of Process (1) in the material, the O_1s_ binding energy will increase due to the effective oxygen oxidation state decrease caused by a deficiency of Zn cations located in regular positions, and an increase in the interaction of the nucleus and electrons located at the core 1s level. Similarly, the O_1s_ binding energy will decrease with the predominance of Processes (2) and (3) in ZnO.

To analyze the obtained results, we consider the change in the binding energy of Zn_2p3/2_ and O_1s_ along with the change in the surface stoichiometry of the material Zn(*lat*.)/O(*lat*.), as well as Zn_2p3/2_ line width at half height (W_Zn_) for each processing type (Table 2). A mechanical 5 h milling of ZnO powders leads to an increase in the binding energy of Zn_2p3/2_ and O_1s_ with a simultaneous decrease in Zn(*lat*.)/O(*lat*.). This indicates the formation of vacancies in the near-surface region of Zn sublattice according to Mechanism (1) with subsequent diffusion of Zn_i_ into the material bulk or capturing by grinding balls, which is accompanied by the depletion of Zn(*lat*.) in the near-surface layer. A slight decrease in the W_Zn_ value observed during milling indicates the energy level alignment of Zn, which occupies different positions in the crystal lattice. This is probably due to both mechanical stress-induced alignment in the crystallites due to milling, and the increase in the powder monodispersity.

Since the time interval between the powder processing and the XPS spectrum measurement was significant (about 2 weeks), we have not analyzed the change in the surface oxygen (O(*ads*.)) concentration due to numerous random factors affecting its state.

Irradiation of commercial ZnO powders with e-beam energy of 0.9 MeV at 1 kGy dosage leads to a simultaneous decrease in the binding energy of Zn_2p3/2_ and O_1s_, and a decrease in Zn(*lat*.)/O(*lat*.), i.e., the depletion of Zn in the near-surface layer. The same case can be observed with a combination of the following processes in ZnO powder: breaking of Zn–O bonds and the transition of zinc and oxygen atoms into interstices; diffusion of Zn_i_ from the near-surface region into the bulk; release of oxygen from a regular position at the crystal lattice nodes (preferred orientation) or interstices into the gas phase.

Such a process can be symbolically divided into several stages, being described by the following quasi-chemical equations:ZnO → Zn_i_ + O_i_ + V_Zn_ + V_O_,(4)
O_O_ → ½O_2_(*gas*)↑,(5)
O_i_ → ½O_2_(*gas*)↑,(6)
where O_i_ is the interstitial oxygen atom. At the first stage, there is breaking of Zn–O bonds, initiated by the electron flow, and release of zinc and oxygen from regular positions thereof in the crystal lattice due to the flying particle energy. In this case, atoms of Zn_i_ diffuse from the near-surface layer into the bulk, while most of the O_i_ remains on the surface. In parallel, there is Process (5) due to potential oxygen release from the surface according to Equation (6). It should be noted that the number of zinc atoms diffusing into the bulk from the near-surface layer exceeds the number of oxygen atoms that have gone into the gas phase in accordance with the change in the surface stoichiometry. We have assumed that simultaneous release of zinc cations and oxygen anions into interstices is not accompanied by a significant change in the binding energies of Zn_2p3/2_ and O_1s_.

The decrease in Zn_2p3/2_ line width at half height from 1.86 eV to 1.80 eV may be related to energy level alignment of Zn in the near-surface layer due to mechanical stress-induced alignment. Evidently, this is due to a smaller number of interstitial defects in the test sample compared to the initial powder.

Irradiation of ZnO powders milled in the attritor for 5 h with e-beam energy of 0.9 MeV at 1 MGy dosage has a number of characteristic features versus irradiation of commercial powders: despite a simultaneous decrease in the binding energy of Zn_2p3/2_ and O_1s_, the ratio of Zn(*lat*.)/O(*lat*.) increases, which indicates Zn influx from the bulk to the surface of the powders. These powders are characterized by the presence of high-energy components in the XPS spectra of Zn_2p1/2_ and Zn_2p3/2_ with binding energy of 1048.2 eV and 1024.9 eV, respectively (Figure 3c), which evidences Zn_i_ atoms [19]. We consider the processes occurring in dispersed powders having been e-beam irradiated.

As stated earlier, the surface of dispersed powders is enriched with vacancies in Zn sublattice before irradiation, and some Zn_i_ atoms are diffused from the surface into the powder bulk at a certain depth. In this regard, the diffusion of Zn_i_, formed during e-beam irradiation of milled powders based on a mechanism similar to that described by Equation (4), from the near-surface layer into the material bulk is hindered due to the presence of a “barrier layer” of Zn_i_, which had been previously formed during milling and occupied the interstices of crystal lattice at some depth. Herein, the occurrence of processes similar to those described by Equations (5) and (6) is probably facilitated in comparison with irradiated commercial powders due to the presence of vacancies in Zn sublattice on the pre-milled surface. All this ultimately leads to the accumulation of Zn_i_ in the near-surface layer, which is recorded by XPS.

A significant increase in Zn_2p3/2_ full width at half maximum (FWHM) of the line confirms the hypothesis of the formation of an excess amount of Zn_i_ in the near-surface region during e-beam processing of milled samples from 1.86 eV to 1.92 eV. This type of defect leads to local pushing apart of the crystal lattice, which results in point stresses and the appearance of nonequivalent Zn^2+^ cations, as opposed to the strain-free lattice. Ultimately, this causes an increase in the value of W_Zn_.

Thus, it can be seen from obtained results that e-beam irradiation of milled samples leads to “removal” of the dispersion effect on the ZnO surface: the binding energies of Zn_2p3/2_ and O_1s_ for samples 1 and 4 are practically the same, and the stoichiometry of the near-surface layer of the samples is partially restored after irradiation. However, the defective structures of the initial surface 1 and surface 4, having undergone both types of processing, are not identical. The close binding energy values indicate that one zinc cation accounts for approximately the same amount of oxygen anions occupying regular positions in the crystal lattice on the surface of samples 1 and 4. This case is observed with the simultaneous release of zinc and oxygen from the nodes as a result of a combination of various types of processing. Herewith, part of Zn_i_ diffuses into the bulk of the powder, and part of the oxygen goes into the gas phase.

We also consider the effect of vacuum heat treatment on the initial samples and those dispersed for 5 h. Figure 4 shows the dependences of the binding energy of Zn_2p3/2_ and O_1s_ (Figure 4a), and the ratios of Zn(*lat*.)/O(*lat*.) and Zn(*lat*.)/O(*ads*.) (Figure 4b). The graphs are given for the initial ZnO powders and those milled for 5 h at the annealing temperatures in the range of 200–500 °C. We have not provided the initial XPS spectra, since they are similar to those shown in Figure 3.

Having analyzed the obtained results, we distinguish several sample heating temperature ranges, each inducing various surface processes. It can be seen from Figure 4 that low-temperature annealing (200–300 °C) has little effect on both the binding energies of Zn_2p3/2_ and O_1s_ and the sample stoichiometry. There is just an insignificant release of both adsorbed and lattice oxygen from the surface, which leads to some increase in Zn(*lat*.)/O(*lat*.) and Zn(*lat*.)/O(*ads*.) for all samples.

A dramatic change in the surface properties was due to the annealing of powders at a temperature of 400 °C. On the one hand, there is a simultaneous increase in both the binding energies of Zn_2p3/2_, O_1s_ and the ratio of Zn(*lat*.)/O(*lat*.) characterizing the surface stoichiometry for the initial powders not subjected to mechanical milling. On the other hand, there is a decrease in the binding energy of Zn_2p3/2_ and O_1s_ with the same stoichiometry for the samples dispersed for 5 h. It should also be noted that a deficiency of oxygen anions is observed on the surface of the initial powders, and a deficiency of zinc cations is detected on the surface of mechanically activated powders. In this case, the binding energies of both O_1s_ and Zn_2p3/2_ for either type of samples become similar, and the surface of a mechanically activated sample turns out to be stoichiometric. The following key findings are based on the obtained results:

(1) the ratio of zinc and oxygen ions, occupying a regular position in the crystal lattice, is similar for both the initial and mechanically activated samples due to the annealing at 400 °C;

(2) a decrease in the binding energy of Zn_2p3/2_ in a mechanically activated sample is associated with oxygen release from regular positions into vacuum, which leads to a decrease in the effective oxidation state of zinc cations;

(3) a simultaneous increase in the binding energies of Zn_2p3/2_ and O_1s_, and the ratio of Zn(*lat*.)/O(*lat*.), indicates oxygen release from regular positions of the crystal lattice into vacuum with simultaneous transition of zinc cations to the interstices (the latter process is predominant). Evidently, such processes can be only observed on a surface with zinc excess and hypothetical transit across the boundary of region homogeneity. In this case, zinc and oxygen occupy their regular positions in almost equal proportions.

Heating of samples at a high temperature (500 °C) is not accompanied by a significant change in the binding energies of Zn_2p3/2_ and O_1s_. Herein, there is a decrease in the stoichiometric ratio of Zn(*lat*.)/O(*lat*.) for both types of samples, and prolongation of adsorbed oxygen release from the surface. This case can be observed due to diffusion of zinc ions from the material bulk into the near-surface layer along the interstitial sites and the absence of a further Zn_i_ → Zn_Zn_ transition. The effect of ZnO surface enrichment with zinc ions during annealing is well known and has been reported, for example, in Ref. [20].

## 4. Conclusions

Using X-ray photoelectron spectroscopy, we have studied the processes in the surface layers of initial and mechanically activated ZnO powders, which occur during e-beam irradiation with energy of 0.9 MeV at 1 MGy dosage, and thermal annealing. It has been shown that a combination of these processes allows fine-tuning of the surface properties (stoichiometry, vacancies, and interstitial atoms), being vital in adsorption, gas sensors, photocatalysis, etc. Using a scanning electron microscopy and X-ray diffraction technique, we have found a decrease in the size of crystallites (from 37 to 21 nm) with a simultaneous increase in microdeformations (from 0.3 to 0.6%) as a result of 5h of milling of commercial powders. Neither change has been established in scanning electron microscopy patterns and diffractograms of the samples due to e-beam exposure thereof. In general, mechanical processing and e-beam irradiation lead to various defective structure characteristics of the near-surface layers. There is a release of zinc cations into the interstices with subsequent diffusion into the material bulk during dispersion. This process is leveled by the release of oxygen anions from regular positions to interstices and the gas phase during irradiation. When mechanically activated samples are irradiated with an e-beam for 5 h, there is a reverse process of zinc influx into the near-surface region from the bulk, but mainly in the form of Zn_i_, being easily detected using X-ray photoelectron spectroscopy. Annealing at a temperature below 400 °C is found to have neither effect on the surface state of the samples. When heating at 400 °C, the XPS spectra of both commercial and mechanically activated samples become practically identical, and a further increase in the heating temperature leads to diffusion of interstitial zinc ions into the near-surface layers of the material bulk.

## Figures and Tables

**Figure 1 nanomaterials-12-01924-f001:**
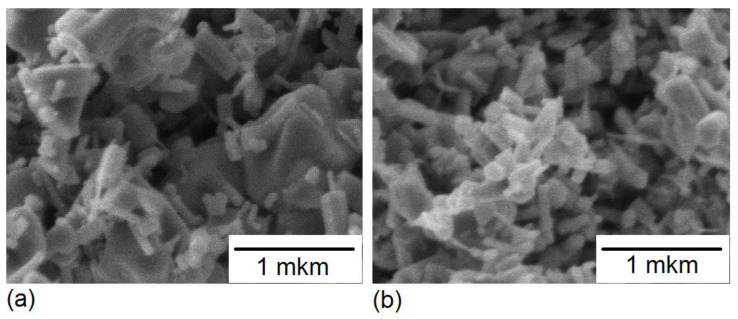
SEM micrographs of ZnO powder: (**a**) commercial and (**b**) mechanically activated for 5 h.

**Figure 2 nanomaterials-12-01924-f002:**
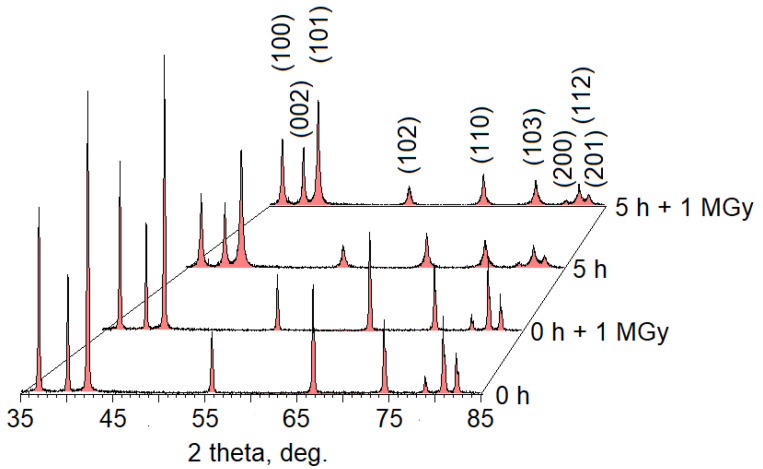
Diffraction patterns of the samples studied.

**Figure 3 nanomaterials-12-01924-f003:**
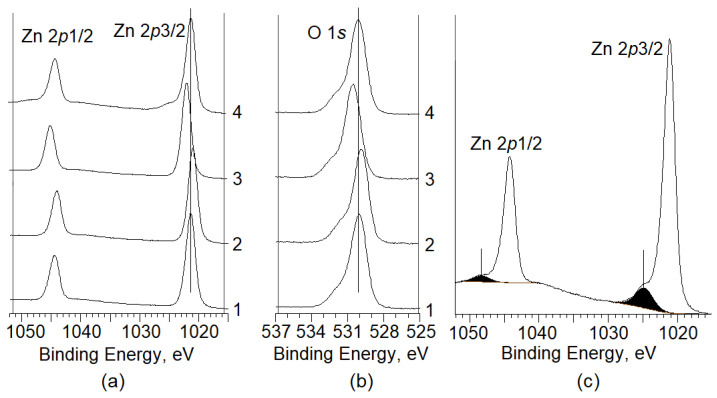
XPS spectra of Zn_2p_ (**a**) and O_1s_ (**b**) for all material processing types; XPS Zn_2p_ spectrum of milled and e-beam-irradiated ZnO powder (**c**).

**Figure 4 nanomaterials-12-01924-f004:**
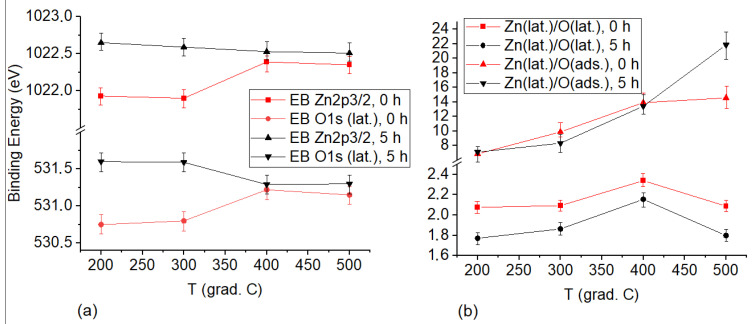
Dependences of the binding energy of Zn_2p3/2_ and O_1s_ (**a**), and the ratio of Zn(*lat*.)/O(*lat*.) and Zn(*lat*.)/O(*ads*.) (**b**) for the initial ZnO powders and those milled for 5 h at different annealing temperatures.

**Table 1 nanomaterials-12-01924-t001:** Binding energies of Zn_2p3/2_, O_1s_, and C_1s_ for all types of powder processing.

Sample	*BE* (Zn_2p3/2_)	*BE* (O_1s_)	*BE* (C_1s_)
1	1021.35 ± 0.1 eV	530.05 ± 0.1 eV	284.80 ± 0.1 eV
2	1021.05 ± 0.1 eV	529.90 ± 0.1 eV	284.80 ± 0.1 eV
3	1022.10 ± 0.1 eV	530.55 ± 0.1 eV	284.80 ± 0.1 eV
4	1021.38 ± 0.1 eV	530.13 ± 0.1 eV	284.80 ± 0.1 eV

**Table 2 nanomaterials-12-01924-t002:** Values of Zn(*lat*.)/O(*lat*.) and Zn_2p3/2_ line width at half height (*W_Zn_*).

Sample	Zn(*lat.*)/O(*lat.*)	*W_Zn_*, eV
1	2.2 ± 15%	1.86 ± 0.01 eV
2	1.6 ± 15%	1.80 ± 0.01 eV
3	1.8 ± 15%	1.85 ± 0.01 eV
4	1.9 ± 15%	1.92 ± 0.01 eV

## Data Availability

Not applicable.

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
