# Peer review of "Control over the Surface Properties of Zinc Oxide Powders via Combining Mechanical, Electron Beam, and Thermal Processing"

_nanomaterials, 2022, doi:10.3390/nano12111924_

Round 1

Reviewer 1 Report

The paper is devoted to a study of defects creation in ZnO powders under combining effect of mechanical activation, electron beam irradiation, and vacuum annealing.
The reported results are interesting, however, some points should be clarified before acceptance as well as the figures quality should be improved. Therefore, I recommend the major revision.  

The XRD patterns should be shown, and the methods used to obtain the size of coherent scattering regions and an estimation of microdeformations should be explained. The errors for the obtained values should be reported. 

A correlation between shifts of the Zn2p and O1s XPS peaks for four samples is observed in Fig. 2: the peaks are shifted to the smaller 
and larger energies for samples 2 and 3, respectively, relative to the samples 1 and 4. How can it be explained? Could it be just a problem with the energy scale determination?
ZnO is an insulator: how was the problem with sample charging during XPS measurements treated?  

The meaning/origin of vertical bars in Fig. 3 is not clear and should be explained in detail. 
I suggest to substitute Fig. 3 with a table reporting binding energies for all samples - it will be more understandable for the reader. 

The numerical values in Table 1 should be reported with errors. 
Error bars are also missing in Fig. 5.

There is no need for a separate Fig. 4: it can be combined with Fig. 2, since the spectrum is 
already reported there.  

The quality of all figures should be improved. They should be prepared using a similar template (axes, letter and number sizes, etc). 
Use a decimal point instead of a comma.

The authors should check the manuscript for typos. For example, in section 2.2, the SEM title should be "VEGA3 SBH" but not "VRGA 3 SBH". 

Author Response

Answers to the reviewers.

We are grateful to the reviewers for their time and work and for their comments and suggestions. We are sure that the improved version of the manuscript, according to the reviewer's suggestions, will be more understandable to the reader.

Answers to the reviewer  1.

  1. The XRD patterns should be shown, and the methods used to obtain the size of coherent scattering regions and an estimation of microdeformations should be explained. The errors for the obtained values should be reported.

Answer:

We have presented XRD patterns in Figure 2. The sizes of the coherent scattering regions and microstrains were determined as follows. The broadening of the diffractogram reflection at half its height B is a mathematical convolution of the broadening associated with the polycrystalline state of the sample (coherent scattering regions) BD and with the broadening associated with the presence of microdeformations BS. These values are given by the following formulas. ​​BD = Kλ/(DcosΘ), BS = 4ηtg(Θ), where K is the form factor; λ is the wavelength of X-ray radiation; D is the size of the coherent scattering regions; Θ is the diffraction angle; η = Δa/a is the value of microstrains, defined as a relative change in the lattice period a.

If one consider a reflex profile as a Cauchy distribution, then B = BD + BS. Analyzing the broadening of several reflections (in principle, two are sufficient), D and η may be calculated. This procedure is discussed in more detail in [Pronin I. A. et al. Investigation of milling processes of semiconductor zinc oxide nanostructured powders by X-ray phase analysis // Journal of Physics: Conference Series. – IOP Publishing, 2017. – V. 917. – no. 3. - P. 032019.]. We have added this brief description to the text of the article (pages 2,3, lines 86-97), and also added inaccuracy values.

  1. A correlation between shifts of the Zn2p and O1s XPS peaks for four samples is observed in Fig. 2: the peaks are shifted to the smaller and larger energies for samples 2 and 3, respectively, relative to the samples 1 and 4. How can it be explained? Could it be just a problem with the energy scale determination? ZnO is an insulator: how was the problem with sample charging during XPS measurements treated?

Answer:

We agrue that the chemical shifts observed are related to changes of the properties of the materials (those changes are described in text page 5, lines 157-176), and not by charging the sample. First, the energy scale of the spectrometer was calibrated using the sputter-cleaned Au surface as a reference, so that the binding energy of the Au4f7/2 peak was set to 84.0 eV. Secondly, as can be seen from Table 1 (we transformed the figure 3 of the earlier version of the manuscript into Table 1), the C1s binding energy in all samples studied was unchanged, 284.8 eV.

  1. The meaning/origin of vertical bars in Fig. 3 is not clear and should be explained in detail. I suggest to substitute Fig. 3 with a table reporting binding energies for all samples - it will be more understandable for the reader.

Answer:

Yes, done. We transformed the Fig.3 into Table 1 reporting binding energies for all samples, according to the reviewers suggestion.

  1. The numerical values in Table 1 should be reported with errors. 
    Error bars are also missing in Fig. 5.

Answer:

Yes, done. We introduced the necessary correction into the text and fig. 5.

  1. There is no need for a separate Fig. 4: it can be combined with Fig. 2, since the spectrum is already reported there.

Answer:

Yes, done. Fig. 4 and fig.,2 are combined, according to the reviewers suggestion.

  1. The quality of all figures should be improved. They should be prepared using a similar template (axes, letter and number sizes, etc). Use a decimal point instead of a comma.

Answer:

Yes, done. Corrections are introduced to figures.

  1. The authors should check the manuscript for typos. For example, in section 2.2, the SEM title should be "VEGA3 SBH" but not "VRGA 3 SBH". 

Answer:

Yes, done. Typos are corrected.

Best regards. We thank the reviewers,

The authors.

Reviewer 2 Report

In the manuscript, the authors adopt X-ray photoelectron spectroscopy to investigate the surface properties of ZnO powders prepared using mechanical, electron beam, and thermal processing. The results may attract interest from the reader of Nanomaterials.

1. In Section 3.1, the author declared that "an increase in microdeformations from 0.3 to 0.6% during 5h milling". How to define and how to measure microdeformations, and what does the change from 0.3 to 0.6 represent?

2. In Section 3.2, the author declared that "In this case, the key process related to an increase in the binding energy is the transition of Zn cations from a regular position (ZnZn) to an interstitial site (Zni) with the formation of Zn  vacancy (VZn) [18]".Are there any other possibilities? In the milled sample, the particles size is smaller, thus there are more surficial atoms with dangling bonds. Does it relate to the shift of binding energy.

3. In Section 3.2, the author declared that "In case of Zni diffusion into the material bulk, there is a change in the surface stoichiometry with a decrease in the value of [Zn]/[O]". What drives the Zni to diffuse into the material bulk during milling. It seems more reasonable to diffuse outwards to the surface.

Author Response

Answers to the reviewers.

We are grateful to the reviewers for their time and work and for their comments and suggestions. We are sure that the improved version of the manuscript, according to the reviewer's suggestions, will be more understandable to the reader.

Answers to the reviewer  2.

  1. In Section 3.1, the author declared that "an increase in microdeformations from 0.3 to 0.6% during 5h milling". How to define and how to measure microdeformations, and what does the change from 0.3 to 0.6 represent?

Answer:

The sizes of the coherent scattering regions and microstrains were determined as follows. The broadening of the diffractogram reflection at half its height B is a mathematical convolution of the broadening associated with the polycrystalline state of the sample (coherent scattering regions) BD and with the broadening associated with the presence of microdeformations BS. These values are given by the following formulas. ​​BD = Kλ/(DcosΘ), BS = 4ηtg(Θ), where K is the form factor; λ is the wavelength of X-ray radiation; D is the size of the coherent scattering regions; Θ is the diffraction angle; η = Δa/a is the value of microstrains, defined as a relative change in the lattice period a.

If one consider a reflex profile as a Cauchy distribution, then B = BD + BS. Analyzing the broadening of several reflections (in principle, two are sufficient), D and η may be calculated. This procedure is discussed in more detail in [Pronin I. A. et al. Investigation of milling processes of semiconductor zinc oxide nanostructured powders by X-ray phase analysis // Journal of Physics: Conference Series. – IOP Publishing, 2017. – V. 917. – no. 3. - P. 032019.]. We have added this brief description to the text of the article (pages 2,3, lines 86-97), and also added inaccuracy values.

  1. In Section 3.2, the author declared that "In this case, the key process related to an increase in the binding energy is the transition of Zn cations from a regular position (ZnZn) to an interstitial site (Zni) with the formation of Zn  vacancy (VZn) [18]".Are there any other possibilities? In the milled sample, the particles size is smaller, thus there are more surficial atoms with dangling bonds. Does it relate to the shift of binding energy.

Answer:

 We argue that the transition of zinc cations from regular positions to interstices is the main mechanism responsible for the increase in the binding energy of Zn 2p3/2. In principle, a mechanism for filling vacancies in the oxygen sublattice (for example, from the environment) with O2– anions is also possible. Nevertheless, the analysis of the atomic composition, of the stoichiometry of the samples, indicates that the contribution of this mechanism is extremely small. Particularly, a noticeable increase in the number of atoms with dangling bonds during milling would be accompanied by a broadening of the Zn 2p3/2 core level peak. However, this does not happen. According to the data in Table 1, the broadening for the initial powder is 1.86 eV, and for the powder milled for 5 hours, it is 1.85 eV. In this regard, we conclude that the contribution of surficial atoms with dangling bonds to the change in the binding energy is insignificant.

  1. In Section 3.2, the author declared that "In case of Zni diffusion into the material bulk, there is a change in the surface stoichiometry with a decrease in the value of [Zn]/[O]". What drives the Zni to diffuse into the material bulk during milling. It seems more reasonable to diffuse outwards to the surface.

Answer:

We agree with the opinion of the reviewer. Although the diffusion of zinc into the bulk of the material cannot be completely ruled out. The removal of zinc from the surface is also possible due to the capture of Zn2+ cations by grinding media made of ZrO2. We introduced the necessary comment in the text of the manuscript (lines 162-163, page 5).

Best regards. We thank the reviewers,

The authors.

Round 2

Reviewer 1 Report

The revision is satisfactory.

Author Response

We thank the reviewer!
Sincerely, authors.

Reviewer 2 Report

The authors have answered my comment and I recommend this manuscript for publication in Nanomaterials.

Author Response

(The authors gave the same response as above.)
